# Arsinothricin Inhibits *Plasmodium falciparum* Proliferation in Blood and Blocks Parasite Transmission to Mosquitoes

**DOI:** 10.3390/microorganisms11051195

**Published:** 2023-05-03

**Authors:** Masafumi Yoshinaga, Guodong Niu, Kunie Yoshinaga-Sakurai, Venkadesh S. Nadar, Xiaohong Wang, Barry P. Rosen, Jun Li

**Affiliations:** 1Department of Cellular Biology and Pharmacology, Herbert Wertheim College of Medicine, Florida International University, 11200 SW 8th St., Miami, FL 33199, USA; 2Department of Biological Sciences, College of Arts, Sciences & Education, Florida International University, 11200 SW 8th St., Miami, FL 33199, USA; 3Biomolecular Sciences Institute, Florida International University, Miami, FL 33199, USA

**Keywords:** arsinothricin, antimalarial, glutamine synthetase, multiple-stage drugs, novel drugs, low toxic drug, malaria, mosquito

## Abstract

Malaria, caused by *Plasmodium* protozoal parasites, remains a leading cause of morbidity and mortality. The *Plasmodium* parasite has a complex life cycle, with asexual and sexual forms in humans and *Anopheles* mosquitoes. Most antimalarials target only the symptomatic asexual blood stage. However, to ensure malaria eradication, new drugs with efficacy at multiple stages of the life cycle are necessary. We previously demonstrated that arsinothricin (AST), a newly discovered organoarsenical natural product, is a potent broad-spectrum antibiotic that inhibits the growth of various prokaryotic pathogens. Here, we report that AST is an effective multi-stage antimalarial. AST is a nonproteinogenic amino acid analog of glutamate that inhibits prokaryotic glutamine synthetase (GS). Phylogenetic analysis shows that *Plasmodium* GS, which is expressed throughout all stages of the parasite life cycle, is more closely related to prokaryotic GS than eukaryotic GS. AST potently inhibits *Plasmodium* GS, while it is less effective on human GS. Notably, AST effectively inhibits both *Plasmodium* erythrocytic proliferation and parasite transmission to mosquitoes. In contrast, AST is relatively nontoxic to a number of human cell lines, suggesting that AST is selective against malaria pathogens, with little negative effect on the human host. We propose that AST is a promising lead compound for developing a new class of multi-stage antimalarials.

## 1. Introduction

Malaria is one of the most severe vector-borne diseases. It is estimated to have caused 241 million new cases and 627,000 deaths in 2020 worldwide [1]. Among the five *Plasmodium* protozoal parasites that infect humans, *P. falciparum* is the deadliest species accountable for most malaria morbidity and mortality globally. Even though a malaria vaccine (RTS,S/AS01) has recently been recommended by the World Health Organization (WHO) [2], vector control is still one of the major methods against mosquito-transmitted diseases. Vector control relies heavily upon chemical insecticides against adult mosquitoes, mostly pyrethrin and its synthetic derivatives, pyrethroids. Because no new classes of insecticides have been put onto the market in nearly 30 years, multiple primary mosquito vectors have developed resistance to pyrethroid insecticides [3,4]. In addition, the rapid spread of resistant parasites to the frontline antimalarials (e.g., artemisinin) emphasizes the need for new antimalarials [5,6]. Passage through the anopheline mosquito is an obligatory step in the malaria parasite life cycle. Thus, blocking *Plasmodium* infection in mosquitoes will halt malaria transmission. Natural products and natural product structures have been the basis of the majority of current antimalarials, playing a significant role in the history of antimalarial drug discovery and development [7,8,9]. However, only a handful of natural compounds have been discovered that inhibit transmission in mosquitoes or have efficacy against multiple stages [10,11,12,13,14,15]. Developing new potent multi-stage drugs is imperative to ensure malaria elimination and eradication.

Arsinothricin (2-amino-4-(hydroxymethylarsinoyl)butanoate, or AST), a novel arsenic-containing natural product synthesized by the rice rhizosphere bacterium *Burkholderia gladioli* GSRB05, is a nonproteinogenic amino acid analog of glutamate that inhibits prokaryotic glutamine synthetase (GS), or type-I GS (GS-I) [16,17]. We previously demonstrated that AST is a potent broad-spectrum antibiotic that effectively inhibits the growth of various bacterial pathogens, including *Mycobacterium bovis* BCG [17], which is closely related to *M. tuberculosis*, the causative agent of human tuberculosis, the top global infectious disease killer. AST is also effective against carbapenem-resistant *Enterobacter cloacae* [17]. Bacteria resistant to this “last resort antibiotic” belong to the highest priority category in the WHO global list of priority pathogens [18]. 

Arsenic has been used in medicine since the eras of Ancient Greece and China [19,20]. In the modern era, Paul Ehrlich chemically synthesized his “magic bullet”, the aromatic organoarsenical salvarsan, the first effective synthetic drug against syphilis. It was the world’s first blockbuster drug and remained the treatment of choice for syphilis until the advent of penicillin in the 1940s. Arsenicals, including organoarsenical melarsoprol, are still in limited use for treating second-stage *Trypanosoma brucei* sleeping sickness [21]. Although no longer in use in the United States, aromatic arsenicals, such as roxarsone (4-hydroxy-3-nitrophenylarsenate) and nitarsone (4-nitrophenylarsenate), have been utilized as antimicrobials for the prevention of *Coccidia* and *Histomonas* infections in poultry [22]. Notably, arsenic trioxide, long used in traditional Chinese medicine, is now a well-established anticancer drug used in combination with all-trans retinoic acid as the treatment of choice for acute promyelocytic leukemia [23]. The success of this arsenic drug for cancer treatment has revived interest in the use of arsenic in medicine—not only as anticancer drugs but also antimicrobials against pathogenic bacteria, parasites and even viruses [19]. New and novel organoarsenicals have the potential to be effective treatments for resistance to antimicrobial agents.

Here, we show that AST is a potent multi-stage antimalarial. Our phylogenetic analysis showed that *Plasmodium* GS is more closely related to prokaryotic GS-I rather than to eukaryotic GS or type-II GS (GS-II), suggesting that AST might be an effective inhibitor of *Plasmodium* GS. As predicted, in vitro assay using purified proteins shows that AST effectively inhibits *P. falciparum* GS-I (PfGS-I) but not human GS-II (hGS-II), suggesting high selectivity of AST for PfGS-I over hGS-II. We further demonstrate that AST possesses a multi-stage antimalarial activity. AST moderately inhibits asexual *P. falciparum* proliferation while it effectively inhibits sexual-stage *P. falciparum* transmission to mosquitoes. In contrast, AST is less cytotoxic to a number of human cell lines, suggesting that AST is highly selective against malaria pathogens with few harmful effects on the human host. Among the tested human cell lines, however, the Caco-2 colon cell line was relatively more sensitive to AST, suggesting potential intestinal toxicity of AST. AST is permeable to various human cell lines, which do not metabolize it. AST is also chemically stable, both extra- and intra-cellularly. Surprisingly, AST is poorly taken up by human erythrocytes, suggesting that AST reduces parasitemia by attacking merozoites released from schizonts rather than inhibiting the proliferation of intracellular parasites. Altogether, our results suggest that AST is a promising lead compound for developing a new class of potent multi-stage antimalarials.

## 2. Materials and Methods

### 2.1. Reagents

Unless otherwise stated, reagents and enzymes were all from MilliporeSigma (Burlington, MA, USA). AST was chromatographically purified from large-scale cultures of *B. gladioli* GSRB05, as described previously [16]. *B. gladioli* GSRB05 was distributed by the NIAS (Institute of Agrobiological Sciences, National Agriculture and Food Research Organization (NARO)) Genebank project (accession number: MAFF211995). The concentration of purified AST was determined using inductively coupled plasma mass spectrometry (ICP-MS) (NexION 1000; Perkin-Elmer, Waltham, MA, USA), while the purity was confirmed through the use of high-pressure liquid chromatography (NexSAR HPLC system, Perkin-Elmer)-coupled to ICP-MS (HPLC-ICP-MS), as described previously [24]. The purified AST is the L-enantiomeric form [17]. Sodium (meta)arsenite, sodium arsenate, disodium methylarsenate and sodium cacodylate were used as arsenite (As(III)), arsenate (As(V)), methylarsenate (MAs(V)) and dimethylarsenate (DMAs(V)). Methylarsenite (MAs(III)) [25], hydroxyarsinothricin (AST-OH) [26], monothio-methylarsenate (T-MAs(V)) [27] and monothio-dimethylarsenate (T-DMAs(V)) [28] were prepared as described previously.

### 2.2. Mosquitos and Plasmodium Strains

Mosquito eggs (G3) were obtained through BEI Resources (Manassas, VA, USA). After hatching in our lab, the mosquito colonies were maintained at 27 °C, as described previously [29], in BSL-2 insectary with a 12 h day-night cycle and 80% humidity. *Plasmodium falciparum* (NF54) parasites were obtained from BEI Resources and maintained in our lab as described previously [30] using fresh O+ human blood and AB+ serum purchased from a blood bank (Continental Blood Bank, Miami, FL, USA). 

### 2.3. Human Cell Lines

Human cell lines from the major tissues/organs, e.g., HepG2 from the liver, HEK293 from the kidney, Caco-2 from the colon, THP-1 from the blood monocyte, and macrophage derived from THP-1 [31], were used to test cytotoxicity, cellular permeability and stability of AST. The cell lines were obtained from the American Type Culture Collection (ATCC) (Manassas, VA, USA). All cells were cultured in their respective media suggested by the vendors in a 5% CO_2_ humidified incubator at 37 °C.

### 2.4. Phylogenetic Analysis of Glutamine Synthetase Homologs

*Escherichia coli* GS type I (NCBI Accession Number: WP_001726606), *Homo sapiens* GS type II (KAI2520623) and *Rozella allomycis* CSF55 GS type III (EPZ33612), representing each of the GS types [32], were used as queries to search for GS homologs using NCBI (National Center for Biotechnology Information) Protein Basic Local Alignment Search Tool (BLASTP) with default settings with a high score threshold of 200. GS homologs were searched in representative parasitic members in major eukaryotic phyla/divisions [33] available in the “Organism” option of BLASTP. Among the identified GS homologs, the following were selected as representatives for phylogenetic analysis: *P. falciparum* NF54 (EWC85923), *P. knowlesi* strain H (XP_038969575), *P. ovale* curtisi (SBS90972), *P. malariae* (XP_028860811), *P. vivax* (XP_001614758), *P. berghei* ANKA (XP_034421207), *Babesia microti* strain RI (XP_012648738), *Toxoplasma gondii* ME49 (XP_002365950), *Eimeria tenella* (XP_013228270), *Cyclospora cayetanensis* (XP_026191025), *Cryptosporidium parvum* Iowa II (XP_627788), *Perkinsus marinus* ATCC 50983 (XP_002779778 and XP_002788110), *Plasmodiophora brassicae* (CEP01483), *Phytophthora kernoviae* 00238/432 (KAF4321638 and KAF4314669), *Blastocystis* sp. ATCC 50177 (OAO13151), *Trypanosoma cruzi* (EKG07062), *T. brucei gambiense* DAL972 (XP_011774949), *Leishmania donovani* (AMQ34913), *L. mexicana* MHOM (XP_003872049), *L. major* strain Friedlin (CAG9568830), *Entamoeba histolytica* KU27 (EMD47977), *Trichomonas vaginalis* G3 (XP_001311020), *Histoplasma capsulatum* H143 (EER45446), *Cryptococcus gattii* WM276 (XP_003191404), *Candida albicans* SC5314 (XP_711992), and *Aspergillus fumigatus* (KAF4291211). In addition to these GS sequences, several more GS homologs were selected from each type [29], as follows: GS-I: *Desulfurococcus amylolyticus* (WP_014766917), *Nostoc punctiforme* (WP_012411650), *Mycobacterium tuberculosis* (WP_003411475), *Bacillus subtilis* (WP_014479846), *Arabidopsis thaliana* (NP_190886), *Oryza sativa* (XP_015614516), *Danio rerio* (NP_001026844), *Ustilago bromivora* (SAM83183), *Thalassiosira pseudonana* CCMP1335 (XP_002288024); GS-II: *Verrucomicrobia* bacterium (RME72265), *Cystobacter fuscus* (WP_095991499), *Arabidopsis thaliana* (NP_188409), *Oryza sativa* (NP_001388898), *Volvox carteri* f. *nagariensis* (XP_002956198), *Drosophila melanogaster* (NP_001162839), *Saccharomyces cerevisiae* S288C (NP_015360), *Thalassiosira pseudonana* CCMP1335 (XP_002294945); GS-III: methanogenic archaeon ISO4-H5 (AMH93813), *Synechococcus* sp. CS-197 (WP_011933362), *Cryobacterium arcticum* (WP_066594742), *Clostridium beijerinckii* (WP_077844654), *Chlorobium phaeobacteroides* (WP_011744711), *Dyadobacter fermentans* (WP_012780063), *Flavobacterium psychrophilum* (WP_011962708), *Ostreococcus lucimarinus* CCE9901 (XP_001415954), and *Thalassiosira pseudonana* CCMP1335 (XP_002295274). Phylogenetic analysis was performed to infer the evolutionary relationship among the sequences of GS from various organisms. The phylogenetic tree was constructed using MEGA XI [34], as illustrated [35], with ClustalW [36] and Neighbor-Joining [37] used for sequence alignment and phylogeny estimation, respectively, where the statistical significance of the branch pattern was estimated from a 1000 bootstrap. Multiple sequence alignment of selected GS was performed using Clustal Omega [38] (https://www.ebi.ac.uk/Tools/msa/clustalo/, accessed on 30 April 2023).

### 2.5. Cloning, Expressing, and Purifying Proteins 

*PfglnA* encoding GS-I from *P. falciparum* NF54 (NCBI accession number: EWC85923.1), a 1626-bp fragment, including the stop codon with codon optimization for expression in *E. coli*, was chemically synthesized by GenScript (Piscataway, NJ, USA) with *Nde*I and *Eco*RI at 5′ and 3′ sites, respectively, and cloned into pMAL-c5x, generating plasmid pMAL-*PfglnA*. *E. coli* BL21(DE3) cells bearing pMAL-*PfglnA* were grown in a lysogeny broth medium [39] supplemented with 25 µM ampicillin at 37 °C with shaking at 200 rpm. At an A_600nm_ of 0.5–0.6, isopropyl ß-D-1-thiogalactopyranoside and glycerol were added at final concentrations of 10 µM and 0.5% (*v*/*v*), respectively. After incubation with shaking at 16 °C overnight, the cells were harvested and stored at −80 °C until use. The frozen cells were thawed and resuspended in buffer A (50 mM morpholinopropane-1-sulfonic acid (MOPS), 1 mM tris(2-carboxyethyl)phosphine, pH 7.5, 0.5 M NaCl and 20% (*v*/*v*) glycerol) (5 mL per g of wet cells). The cells were lysed in a single passage through a French pressure cell at 20,000 psi, and the cell lysate was centrifuged at 40,000 rpm using a T865 rotor (Thermo Fisher Scientific, Waltham, MA, USA) for 60 min at 4 °C. The supernatant solution was applied onto an amylose resin column (New England BioLabs, Ipswich, MA, USA) at a flow rate of 1.0 mL/min and washed until no protein was detected from the flow-through using the method of Bradford. Bound protein was eluted with buffer A containing 10 mM maltose, and the purity was analyzed using SDS-PAGE. The protein concentrations were estimated using the method of Bradford with bovine serum albumin used as a standard. Fractions containing the target protein were pooled and concentrated using a 30 kDa Amicon Ultra centrifugal filter (MilliporeSigma). The concentrated protein was rapidly frozen and stored at −80 °C until use.

### 2.6. Glutamine Synthetase Assays

The activity and inhibition profiles of purified MBP-fused PfGS-I or human GS-II (hGS-II) (BioVision Inc., Milpitas, CA, USA) in the presence or absence of AST were estimated from glutamine production using Glutamine/Glutamate-GLO^TM^ Assay Kit (Promega, Madison, WI, USA). The assay mixture (50 mM imidazole-HCl buffer (pH 7.0), 7.6 mM ATP, 50 mM magnesium chloride, 50 mM ammonium chloride) with or without the indicated concentration of GS enzyme was preincubated in the presence or absence of the indicated concentrations of AST for 15 min at room temperature. The reaction was initiated through the addition of L-glutamate at 0.1 mM, the final concentration. The reaction mixtures were incubated at 37 °C with shaking at 300 rpm for the indicated time. The reactions were terminated by chilling the reaction mixture in ice water and filtered with 30 kDa Amicon Ultra centrifugal filters at 4 °C. Glutamate concentrations in the filtered samples were determined using a Glutamine/Glutamate-GLO^TM^ Assay as instructed in the kit’s manual, from which glutamine concentrations were estimated by subtracting the glutamate concentrations from the initial concentrations (0.1 mM). The EC_50_ (half maximal effective concentration) of AST for PfGS-I was calculated using Sigma Plot (Systat Software, Inc., Sun Jose, CA, USA).

### 2.7. Antimalarial Activity of AST on the Asexual Stage of P. falciparum 

3–5-day cultured *P. falciparum*-infected red blood cells (iRBCs) were mixed with fresh human red blood cells (RBCs) (AB+ type) in complete RPMI 1640 to prepare cultures with 0.5% parasitemia and 2% hematocrit. AST was dissolved in dimethyl sulfoxide (DMSO). Then, 5 μL DMSO with AST at concentrations of 0, 0.5, 1, 2, 4, and 8 mM were added to the 0.5 mL iRBCs in a 48-well plate, yielding final concentrations of 5, 10, 20, 40, and 80 µM, respectively. The plate was incubated in a candle jar at 37 °C. Approximately 48 h later, the medium was replaced with fresh medium containing the same concentration of AST. Parasitemia was recorded via microscopic evaluation of Giemsa-stained blood smears on day 4 post-incubation.

### 2.8. Transmission-Blocking Assays of AST against P. falciparum

As described previously [13,14], *P. falciparum* NF54 was cultured in the complete RPMI 1640 containing 4% fresh O+ human RBCs, 10% human AB+ serum, and 12.5 μg/mL of hypoxanthine in a candle jar at 37 °C. Standard membrane feeding assays (SMFA) [30] were used to examine the efficacy of AST. In brief, 10 mL of day-15 cultured *P. falciparum* containing ~2% stage V gametocytes in a 15 mL centrifuge tube was centrifuged at 650× *g* for 5 min at room temperature. After the supernatant was removed, the cells were resuspended in 1 mL of pre-warmed human AB- serum (37 °C), to which 1 mL of O+ hematocrit was added. In total, 300 μL of the resulting infected blood was mixed with 2 μL of AST at different concentrations, yielding final concentrations of 0, 0.1, 0.3, 1, 3, or 30 μM, which were fed one hundred 3–5-day-old female mosquitoes for 30 min. The unfed mosquitoes were removed, and the engorged mosquitoes were maintained with 10% sugar (pure granulated sugar in distilled water) in the insectary. Seven days post-infection, the midguts were dissected and stained with 0.1% mercury dibromofluorescein salt in phosphate-buffered saline (PBS) for 30 min. The number of oocysts was counted under a 10× light microscope. 

### 2.9. Cytotoxicity Assays

THP-1 cells were seeded in a 24-well plate at a density of 1.0 × 10^5^ cells/well, while the other cell types were seeded at a density of 3.0 × 10^4^ cells/well in 96-well plates. After 24 h, the cells were further cultured in the presence or absence of the indicated concentrations of AST or As(III) for another 72 h, following which viability of cells was determined by a 3-(4,5-dimethylthiazol-2-yl) 2,5-diphenyltetrazolium bromide (MTT) assay [40]. MTT was added to each well at a final concentration of 0.5 mg/mL, and the cultures were incubated for 3 h. The plate of THP-1 cells was then centrifuged at 400× *g*; the cell pellets were lysed with 300 µL of DMSO to dissolve MTT formazan. For the other cell types, the MTT-containing medium was removed from each well, following which 50 µL of DMSO was added to solubilize the formazan. The cell viability was measured using A_570nm._

### 2.10. Membrane Permeability and Stability of AST in Human Cell Lines

The above-mentioned cell types were seeded at the density of 2.0 × 10^6^ cells/well in 12-well plates and incubated with or without 100 µM AST in their respective media for 48 h. The culture media were collected for arsenic speciation. After washing four times with PBS, the cells were collected with cell scrapers or centrifuge (400× *g*) and dried at room temperature for 48 h. For measurement of cellular permeability, the dried cells were digested with 70% nitric acid (≥99.999% trace metals basis) at 70 °C for 60 min, allowed to cool to room temperature and diluted to a final concentration of 2% nitric acid with HPLC-grade water, and the total arsenic content of each sample was quantified using ICP-MS. For analysis of intracellular AST stability, the dried cell pellets were resuspended in 10 mM Tris-NO_3_ (pH 7.4) (2.5 mL per 1 mg of dry cells). The cells were lysed using sonication on ice using a sonic dismembrator (Model 120, Thermo Fisher Scientific) with 20% power in 20 s on–5 s off cycles for a total of 3 min. The cell lysates were centrifuged at 16,200× *g* at 4 °C for 10 min, and the resulting supernatants were filtered with 3 kDa Amicon Ultra centrifugal filters at 4 °C. The collected culture media were also filtered in the same fashion. Arsenic species in the filtered samples were analyzed by HPLC-ICP-MS using a reverse-phase C18 column (BioBasic^TM^ 18 (particle size, 5 µm; length, 250 mm × 4.6 mm)) (ThermoFisher Scientific), as described previously [24]. A standard solution containing As(III), As(V), MAs(III), MAs(V), T-MAs(V), DMAs(V), T-DMAs(V), AST and AST-OH was freshly prepared and run at the beginning and/or end of each batch of sample analysis.

### 2.11. GS-AST Docking

The AST molecule taken from Protein Data Bank (PDB) entry 5WPH [17] was docked with the cryogenic electron microscopy (cryoEM) structure of PfGS-I (PDB ID: 6PEW) [41] using AutoDock4 [42]. The grid center was positioned on the glutamate-binding site of PfGS-I with a dimension of 40 × 40 × 40 Å^3^ with default settings. The top-ranked confirmation was selected for further analysis. The same method was used for hGS-II (PDB ID: 2QC8) [43]. The molecular graphics were drawn using PyMol (Version 1.8, Schrödinger, Inc., New York, NY, USA).

### 2.12. Statistics

Assays of glutamine synthetase, AST antimalarial activity, cytotoxicity, permeability, and stability were repeated at least three times as indicated. Data are presented as the mean ± standard error (SE). An unpaired *t*-test was used to calculate the *p*-values of drug effects on GS activity. One-way ANOVA test and Wilcoxon-Mann–Whitney test were used to calculate the *p*-values of drug effects on malaria infection in blood and malaria oocysts, respectively. The values of EC_50_ were calculated using the Quest Graph™ LC_50_ (median lethal concentration) Calculator from AAT Bioquest Inc. (Pleasanton, CA, USA). 

## 3. Results

### 3.1. Plasmodium GS Is Phylogenetically Closer to Prokaryotic GS-I than to Eukaryotic GS-II

To our knowledge, *Plasmodium* glutamine synthetase has not been examined as a target for antimalarial development. Using PlasmoDB (https://plasmodb.org/plasmo/app/, accessed on 10 September 2021), a functional genomic database for malaria parasites [44], we found that *P. falciparum* 3D7 (a clone of strain NF54) possesses only a single GS gene in its genome (Gene ID: PF3D7_0922600). There are three distinct groups of glutamine synthetases [32]. Type I (GS-I, encoded by *glnA*) and type II (GS-II, *glnII*) are the predominant forms in prokaryotes and eukaryotes, respectively. Type III (GS-III, *glnN*) was recently recognized in a few prokaryotes and eukaryotes. We conducted a phylogenetic analysis of representative parasite GS from seven phyla and one division (Figure 1). GS-II, the predominant form in eukaryotes, is found in three phyla and one division, and GS-III is found in four phyla. In contrast, GS-I, the predominant form in prokaryotes, is also present in the eukaryotic phylum Apicomplexa, including *Plasmodium* species. The single exception is *Perkinsus marinus* in the phylum Perkinsozoa. Our results demonstrate that *Plasmodium* GS belongs to the GS-I family. This is consistent with the recently reported cryoEM structure of *P. falciparum* GS (PfGS) [41], which shows that it is structurally similar to *Salmonella enterica* GS-I, forming a homo-dodecameric complex that adopts a two-tiered ring shape with hexametric symmetry.

### 3.2. PfGS-I Is a Functional GS

Prokaryotic GS-I proteins found in some eukaryotes frequently do not have GS catalytic activity and may have different functions [32]. In humans, for example, most tissues/organs express only catalytically active GS-II. In contrast, human GS-I does not exhibit GS activity. It is expressed only in the lens of the eye and has designated lengsin (lens GS-like protein), probably with a structural role [45]. To examine the catalytic activity of PfGS-I, we cloned and expressed its gene in *E. coli* and assayed the GS activity of the purified protein. PfGS-I produced glutamine from glutamate in the presence of ATP, Mg^2+^ and ammonia (Figure 2a), demonstrating that PfGS-I is a functional GS.

### 3.3. AST Effectively Inhibits PfGS-I but Not Human GS-II

We predicted that AST might be as potent an inhibitor of *Plasmodium* GS-I as it is of *Escherichia coli* GS-I [17] but not human GS-II (hGS-II). We examined the inhibition activity of AST on purified recombinant PfGS-I and compared it with that on hGS-II (Figure 2b). In the presence of 1 µM AST, the activity of PfGS-I (1 µM) was less than 40%, compared with the control without AST treatment. In contrast, treatment with 1 µM AST did not significantly reduce the activity of hGS-II (1 µM), suggesting that AST is much less effective on hGS-II. AST inhibited PfGS-I in a concentration-dependent manner, with an EC_50_ value in the sub-micromolar range (0.4 μM, Figure 2c). Our results demonstrate that AST is an effective PfGS-I inhibitor with high selectivity for PfGS-I over hGS-II. 

### 3.4. AST Inhibits Asexual-Stage P. falciparum Proliferation

Both transcriptomics and proteomics data indicate that PfGS-I is expressed throughout all stages of the malaria parasite life cycle [46,47,48] (https://plasmodb.org/plasmo/app/record/gene/PF3D7_0922600#category:proteomics, accessed on 30 April 2023). We hypothesize that AST might be a potent multi-stage antimalarial drug through the inhibition of PfGS-I. We first examined the effect of AST on *P. falciparum* proliferation in human blood. AST was added into *P. falciparum*-infected blood and measured parasitemia as the percentage of infected cells on day 4. Our results showed that AST inhibits asexual-stage *P. falciparum* proliferation in the blood in a dose-dependent manner (Figure 3). The EC_50_ for AST for inhibition of the asexual-stage *P. falciparum*, defined as the concentration of AST that inhibits 50% of infection intensity (the parasitemia rates (%)) compared to that of the AST-free control, was 13.9 µM. Since most currently available antimalarial drugs and candidate drugs in clinical development have EC_50_ values in the nanomolar range [15], AST has only moderate activity against asexual blood stage forms. 

### 3.5. Transmission Blocking Activity of AST 

We next analyzed the effects of AST on *P. falciparum* transmission to *Anopheles gambiae* mosquitoes. AST was added to 15-day cultured *P. falciparum*-infected blood at concentrations from 0.1 to 3 µM and fed to *An. gambiae* using SMFA. The number of oocysts in mosquito midguts was counted on day 7 post-infection. AST completely inhibited malaria transmission at 3 µM (Figure 4), suggesting that AST effectively blocks transmission. Currently available antimalarial drugs and candidate drugs in clinical development, with a few exceptions, require 5 µM or higher for complete inhibition of *P. falciparum* transmission in SMFAs [15]. These results demonstrate that AST is at least as effective as current drugs. In contrast, no dead mosquitoes were observed, suggesting that AST has no or little insecticidal activity. The EC_50_ of AST in blocking the transmission of the sexual-stage *P. falciparum* to mosquitos, defined as the concentration of a compound that inhibits 50% of infection intensity (the number of oocysts per mosquito) compared to that of the compound-free control, was 0.34 µM.

### 3.6. AST Shows Low Cytotoxicity in Human Cell Lines

The AST cytotoxicity was evaluated and compared with that of inorganic As(III) using five different types of human cell lines from major organs/tissues: HEK293, immortalized embryonic kidney cells; THP-1, monocytes derived from an acute monocytic leukemia patient; macrophage, macrophage-like cells differentiated from THP-1; HepG2, immortalized cells isolated from a hepatocellular carcinoma; and Caco-2, immortalized cell line derived from a colorectal adenocarcinoma patient (Figure 5). The results show that AST has much lower cytotoxicity in human cells than As(III). The LC_50_ values of AST on all the tested cell lines except Caco2 were greater than 250 μΜ. Caco-2 was relatively more sensitive to AST with a lower LC_50_ value (150–200 μΜ), suggesting a potential toxic effect of AST in intestinal tissue. In contrast, the LC_50_ values of As(III) on all the tested cell lines except macrophage were lower than 25 μΜ, while that of macrophage was higher (100 μΜ), suggesting that AST is >10 times less cytotoxic than As(III). AST at 100 µM completely inhibited PfGS-I activity (Figure 2c), *P. falciparum* proliferation in the blood (Figure 3) and transmission to mosquitoes (Figure 4) but had little effect on most of the tested human cell lines (Figure 5). Thus, AST is effective against the malaria parasite with limited effect on human cells. 

### 3.7. AST Permeability and Stability in Human Cells

We investigated the permeability and stability of AST in cells of the five human cell lines. After incubation with AST at 100 μM for 2 days, total arsenic levels in cells (Figure 6) and arsenic species in culture media (Figure 7a) and cell lysates (Figure 7b) were analyzed. Total arsenic levels increased inside all cell lines, yet the levels varied depending on the cell types, with the highest in Caco2 (Figure 6). Arsenic species in both culture media (Figure 7a) and cell lysates (Figure 7b) were almost exclusively AST, demonstrating that AST is permeable and stable in the tested human cell lines.

We also examined AST permeability and stability in human erythrocytes (Figure 7a,b), which is relevant to the antimalarial activity of AST against asexual-stage parasites. *Plasmodium* parasites increase the permeability of the erythrocyte membrane to various low-molecular-weight solutes required for growth by forming a new permeability pathway [49]. The predominant arsenic species in the culture medium was nearly exclusively AST, indicating that erythrocytes do not metabolize AST. Given AST’s moderate anti-asexual activity (Figure 3), we anticipated that the erythrocyte membrane would be permeable to AST. Surprisingly, erythrocytes took up little AST regardless of malaria infection (Figure 6 and Figure 7b). From these results, it appears that AST exerts its effect extracellularly, reducing parasitemia by attacking free merozoites released from schizonts rather than affecting parasites inside erythrocytes. Given that merozoite invasion of erythrocytes occurs within 5–10 min after rupture [50], AST can be effectively taken up by free merozoites, which in turn prevents erythrocyte invasion and/or proliferating, resulting in reduced parasitemia in a concentration-dependent manner.

## 4. Discussion

Drug resistance is a significant factor in the increasingly difficult control of malaria. New drugs against new targets are urgently needed. AST, a newly discovered inhibitor of bacterial GS-I, is an effective antibiotic against various bacterial pathogens [17]. Phylogenetically, *Plasmodium* GS is more closely related to prokaryotic GS-I than to eukaryotic GS-II. Notably, the only active form of human GS (hGS-II), which is widely expressed in the major tissues/organs, belongs to GS-II. Our data clearly show that AST highly selectively inhibits PfGS-I over hGS-II. GS-I, which consists of 12 identical subunits forming two hexameric rings, is evolutionarily and structurally distinct from GS-II, which consists of 10 subunits forming two pentameric rings [51]. Despite the structural differences between the two forms, the residues involved in catalysis [43] are highly conserved in PfGS-I and hGS-II (Figure 8). To shed light on the AST specificity, we conducted docking studies of AST on PfGS-I and hGS-II (Figure 9). The width of the active-site cavity of PfGS-I is approximately 10 Å, which allows the AST molecule to access the active site (Figure 9a). AST molecule coordinates with six residues, including two glutamate-binding residues and one ammonia-binding residue (Figure 9b), preventing the glutamate substrate from entering the active site. In contrast, the width of the active-site cavity in hGS-II is approximately 6.5 Å (Figure 9c), much narrower than that of PfGS-I due to the close interaction of active site loops. The narrow cavity would limit the access of the AST molecule to the active site of hGS-II. We propose that the selectivity of AST for PfGS-I over hGS-II can be attributed to the difference in their active-site cavity size. Co-crystallization of PfGS-I with AST is underway, which will provide further insights into the selectivity. Our data show that AST is highly selective against malaria pathogens with limited effect on human cell lines. Among the tested human cell lines, however, Caco-2 was relatively more sensitive to AST (Figure 5), suggesting a potential toxic effect of AST in intestinal tissue. Caco-2 takes up AST more effectively than the other cell types (Figure 6 and Figure 7b), which we predict leads to partial inhibition of hGS-II, making Caco-2 more vulnerable to AST. Further investigations will be required to assess and address the potential adverse health effects of AST in future study. It is reasonable to hypothesize that the primary mechanism of action of the multi-stage antimalarial AST is inhibition of PfGS-I, which is expressed throughout all stages of the parasite life cycle. 

Eukaryotic protozoic parasites, such as *Leishmania* and *Trypanosoma*, have incomplete or non-functional urea cycles [52]. In these parasites, GS is proposed to play substantial roles in nitrogen metabolism, including in ammonia detoxification and pH regulation [53,54,55,56]. Inhibition or knockout of GS attenuates growth, differentiation and infectivity in those trypanosomatids [53,55,56]. *Plasmodium* also has an incomplete urea cycle, lacking the two major enzymes ornithine transcarbamoylase and argininosuccinate lyase (https://plasmodb.org/plasmo/app/record/pathway/MetaCyc/PWY-4984, accessed 14 October 2022), suggesting that GS might also play a key role in ammonia detoxification, as well as pH homeostasis that is essential for organelles such as digestive vacuoles [57]. The multi-stage antimalarial activity of AST may be attributed to one or a combination of accumulation of toxic ammonia and pH increases via PfGS-I inhibition.

Apicomplexans, including *Plasmodium* species, are obligate intracellular eukaryotic pathogens responsible for a wide range of human and animal diseases. Our phylogenetic analysis shows that *Plasmodium* species and other major Apicomplexans possess prokaryotic type-I GS. According to transcriptomics and proteomics data available in ToxoDB (https://toxodb.org/toxo/app/, accessed on 1 December 2022), the functional genomic resource for *Toxoplasma* and related parasites [58], *Toxoplasma gondii* [59,60,61,62] (https://toxodb.org/toxo/app/record/gene/TGME49_273490#category:proteomics, accessed on 30 April 2023), *Besnoitia bensnoiti* (https://toxodb.org/toxo/app/record/gene/BESB_007290#category:transcriptomics, accessed on 30 April 2023) and *Eimeria tenella/falciformis* [63,64,65], the causative agents for toxoplasmosis, besnoitiosis and coccidiosis, respectively, express GS-I throughout multiple stages of their life cycle. Transcriptomics and proteomics data from the cryptosporidium parasite database CryptoDB (https://cryptodb.org/cryptodb/app/, accessed on 1 December 2022) [66] demonstrate that *Cryptosporidium parvum* [67] (https://cryptodb.org/cryptodb/app/record/gene/cgd6_4570#category:proteomics, accessed on 30 April 2023) also express GS-I throughout multiple stages of their life cycle. This parasite is the causative agent of cryptosporidiosis. Given that all apicomplexan parasites lack at least some components of the urea cycle [68,69], glutamine synthetase may be a potential target to develop multi-stage drugs against not only malaria but also other apicomplexan diseases.

Recently, with high-throughput *piggyBac* transposon insertional mutagenesis and quantitative insertion site sequencing, Zhang et al. generated a saturation-level *P. falciparum* mutant library, distinguishing essential and dispensable genes for in vitro asexual blood-stage growth based on “mutability” [70] (data available in PlasmoDB). Surprisingly, the deletion of *PfglnA*, the gene encoding PfGS-I, has little impact on parasite growth under standard culture conditions the in vitro blood stage, suggesting that the gene is dispensable for asexual development. On the other hand, the growth of *PfglnA* mutants is negatively affected in competitive growth conditions, indicating that the gene plays a beneficial role in growth fitness. Thus, AST may have other target(s) besides PfGS-I. Effective drugs often have multiple targets [71], as represented by the front-line antimalarial artemisinin [72].

Given its poor permeability in erythrocytes, it would be reasonable to consider that AST primarily targets free merozoites released from schizonts rather than parasites inside erythrocytes. Intraerythrocytic gametocytes, the sexual precursor cells of the malaria parasite, are activated in the mosquito midgut and subsequently exit from the enveloping erythrocytes for gamete formation and fertilization [73]. As is the case with the asexual stage parasites, therefore, AST is more likely to act after parasite egress in the sexual stage, killing parasites via GS inhibition. During the asexual stage, parasites replicate inside red blood cells. When merozoites are released after the rupture, they rapidly invade uninfected erythrocytes within 5–10 min to repeat the cycle [50], whereas the sexual stage gametocytes undergo two relatively longer processes of stage conversion, from gametocytes to gametes (~2 h) and from zygotes to ookinetes (~18 h), without the erythrocyte membrane barrier [73]. The shorter time of the exoerythrocytic period might be a major reason why a higher concentration of AST is required to control asexual stage parasites compared to the sexual stage parasites. Development of AST derivatives with higher erythrocyte permeability may lead to more effective multi-stage antimalarials that can directly attack intraerythrocytic parasites. In contrast to RBCs, the other human cell lines are permeable to AST. Elucidation of the mechanisms involved in AST uptake would provide key information to improve AST permeability. 

## Figures and Tables

**Figure 1 microorganisms-11-01195-f001:**
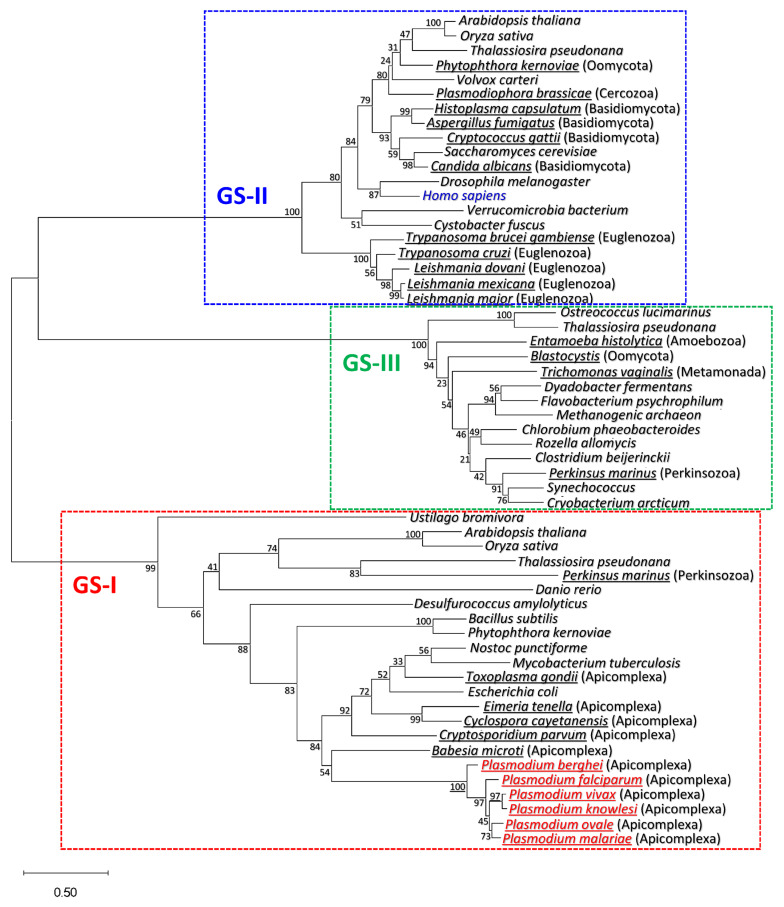
Phylogenetic tree of glutamine synthetases. The neighbor-joining phylogenetic tree shows the evolutionary relationships of glutamine synthetase homologs from representative members of the three distinct groups GS-I, GS-II, and GS-III. GS from parasites are underlined, and their phyla or divisions are given in parentheses. *Plasmodium* species and *Homo sapiens* are highlighted in red and blue, respectively. Bootstrap values calculated for 1000 subsets (%) are indicated on each branch. NCBI accession numbers of proteins are given in *Materials and Methods*. The scale bar represents 50% sequence dissimilarity.

**Figure 2 microorganisms-11-01195-f002:**
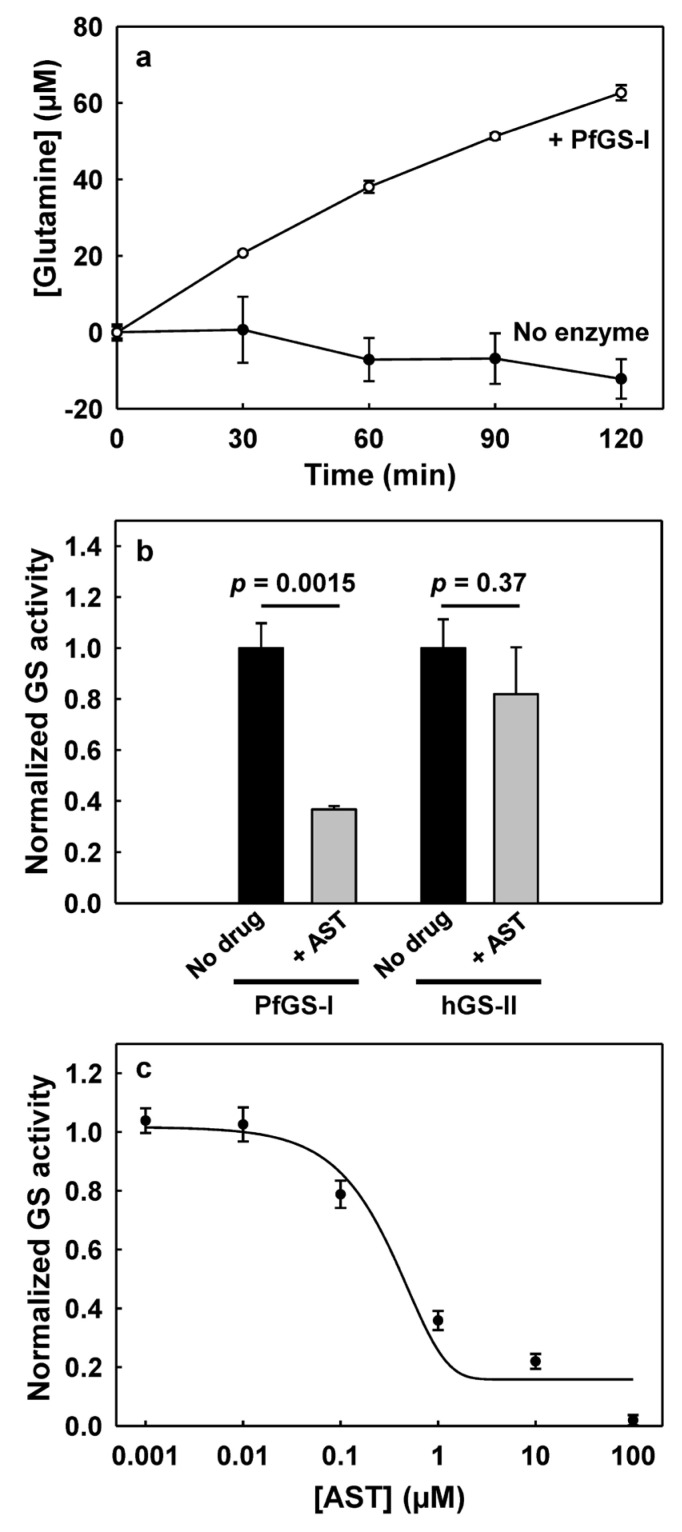
AST inhibits PfGS-I but not human GS-II. (**a**) PfGS-I is a functional glutamine synthetase. The amount of glutamine produced from glutamate in the presence or absence of purified PfGS-I (0.5 µM) was measured every 30 min for 2 h. (**b**) AST effectively inhibits PfGS-I but not human GS-II. The amount of glutamine produced from glutamate by the enzyme (1 µM) was determined in the presence or absence of 1 µM AST in 30 min. (**c**) AST inhibits PfGS-I in a concentration-dependent manner. The activity of PfGS-I (0.5 µM) was assayed in the presence or absence of various concentrations of AST in 2 h. Data are the mean ± SE (n = 3). An unpaired *t*-test was used to calculate the *p*-values. In (**b**,**c**), activities are normalized to the relative activity of untreated control reactions (1.0).

**Figure 3 microorganisms-11-01195-f003:**
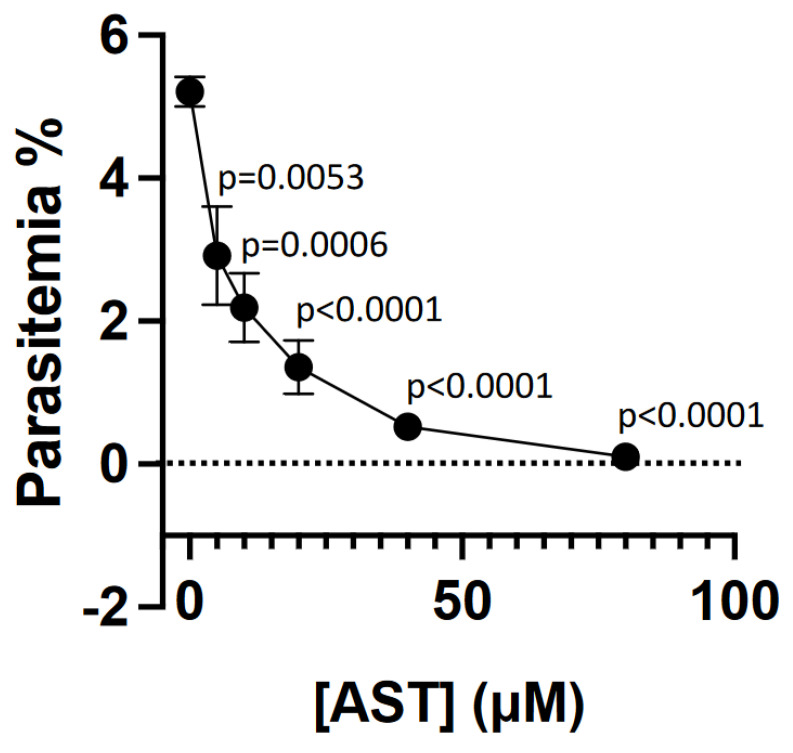
AST inhibits blood-stage *P. falciparum* proliferation. *P. falciparum*-infected erythrocytes with 0.5% parasitemia were incubated at 2% hematocrit in the presence or absence of the indicated concentrations of AST for 4 days, and parasitemia were determined, as described in *Materials and Methods*. Data are the mean ± SE (n = 3). A one-way ANOVA test was used to calculate the *p*-values.

**Figure 4 microorganisms-11-01195-f004:**
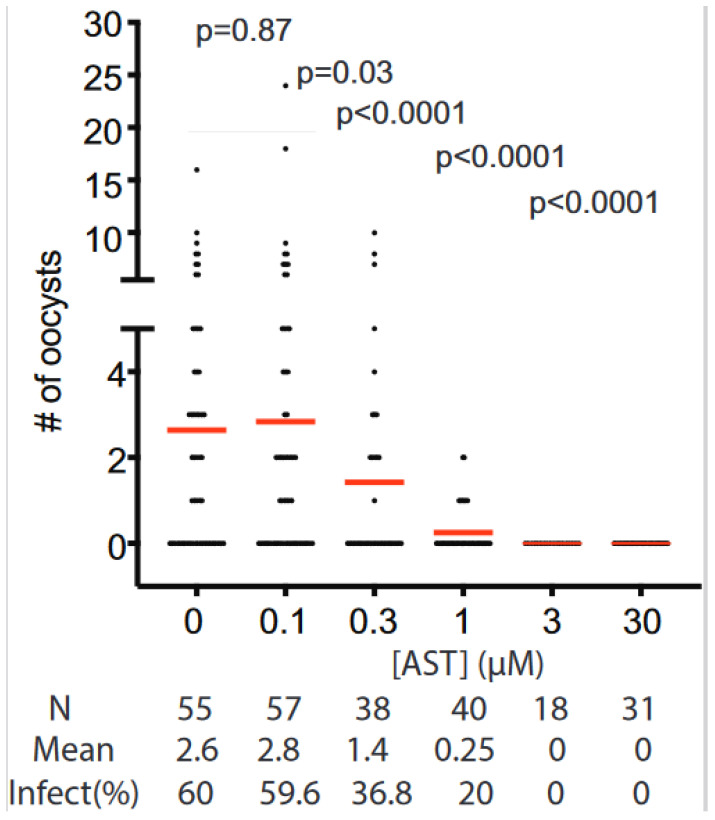
AST blocks *P. falciparum* transmission to *An. gambiae*. The effects of AST on *P. falciparum* infection in *Anopheles gambiae* mosquitoes were analyzed using an SMFA as described in *Materials and Methods*. Each dot on the chart represents a single midgut count. The mean oocyst number per midgut for each data set is indicated with a red horizontal line. N = number of mosquitoes analyzed; Mean = mean oocyst number per midgut; Infect(%) = percentage of infected mosquitoes out of the total number of analyzed mosquitoes. A Wilcoxon-Mann–Whitney test was used to calculate the *p*-values.

**Figure 5 microorganisms-11-01195-f005:**
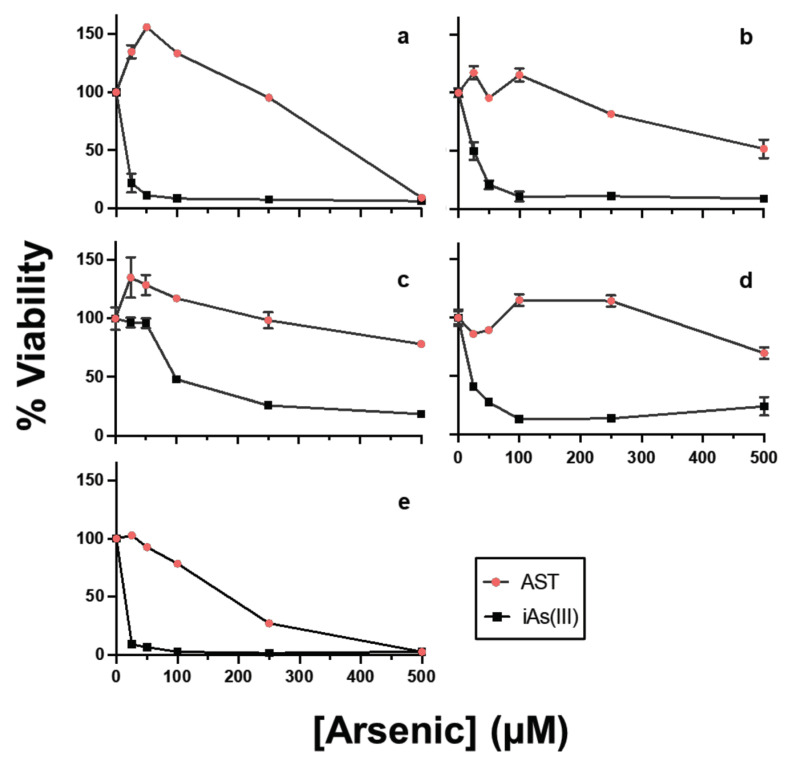
AST is nontoxic to human cell lines. Five different types of human cell lines from major organs/tissues [(**a**) HEK293, immortalized embryonic kidney cells; (**b**) THP-1, monocytes derived from an acute monocytic leukemia patient; (**c**) macrophage, macrophage-like cells differentiated from THP-1; (**d**) HepG2, immortalized cells isolated from a hepatocellular carcinoma; and (**e**) Caco-2, immortalized cell line derived from a colorectal adenocarcinoma patient] were incubated in the presence or absence of the indicated concentrations of AST (red circles) or As(III) (black squares) for 72 h, and viabilities were determined using MTT assays. Viabilities were normalized to the relative viability of untreated control samples (100%). AST cytotoxicity was evaluated and compared with that of As(III) using five different types of human cell lines. Data are presented using mean ± SE (n = 4).

**Figure 6 microorganisms-11-01195-f006:**
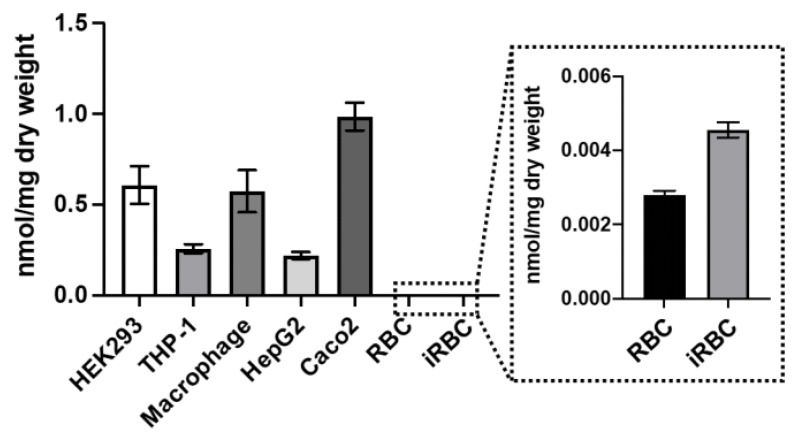
Permeability of AST to various human cell cytoplasmic membranes. Human cell types were incubated with and without 100 µM AST for 48 h, and total cellular arsenic levels were quantified using ICP-MS. Inset: The arsenic levels in RBC and iRBC are in the lower range (up to 0.006 nmol/mg dry weight). The values were adjusted by subtracting control values (cells incubated without AST) from the values of cells incubated with AST. Data are the mean ± SE (n = 3).

**Figure 7 microorganisms-11-01195-f007:**
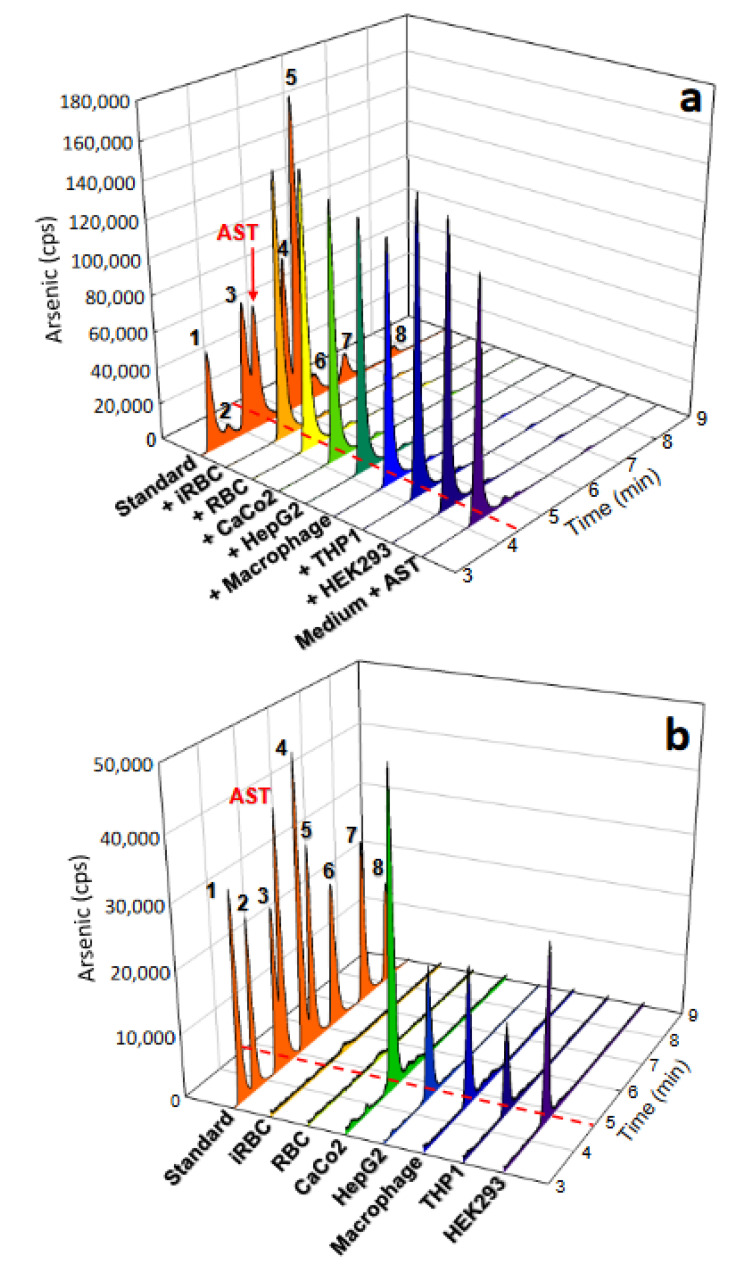
AST is stable in culture media and human cell lines. Human cell lines were incubated with AST at 100 µM for 48 h. The arsenic species in culture media (**a**) and cell lysates (**b**) were analyzed by HPLC-ICP-MS. The data represent three replicates. Standard: 1, As(III); 2, MAs(III); 3, DMAs(V); 4, AST-OH; 5, MAs(V); 6, T-DMAs(V); 7, As(V); 8, T-MAs(V).

**Figure 8 microorganisms-11-01195-f008:**
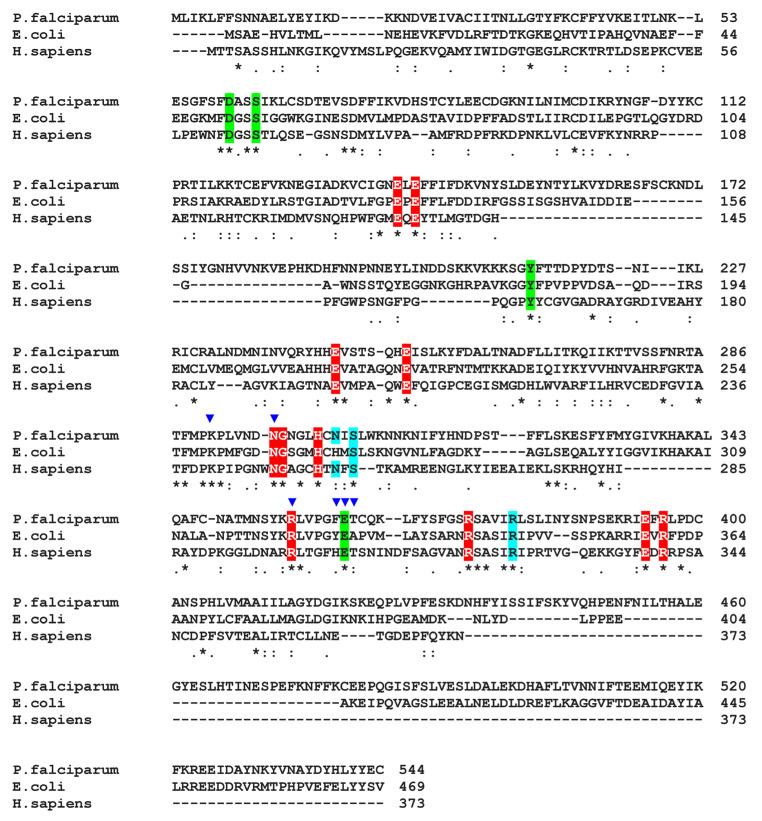
Multiple protein sequence alignment of PfGS-I, EcGS-I and hGS-II. Residues involved in the binding of glutamate (red), ATP (cyan), and ammonia (green) are highlighted [43]. Residues predicted to be involved in the binding of AST are highlighted by blue inverted triangles. Asterisks indicate fully conserved residues. Colons and periods indicate conservations between groups of strongly and weakly similar properties, respectively.

**Figure 9 microorganisms-11-01195-f009:**
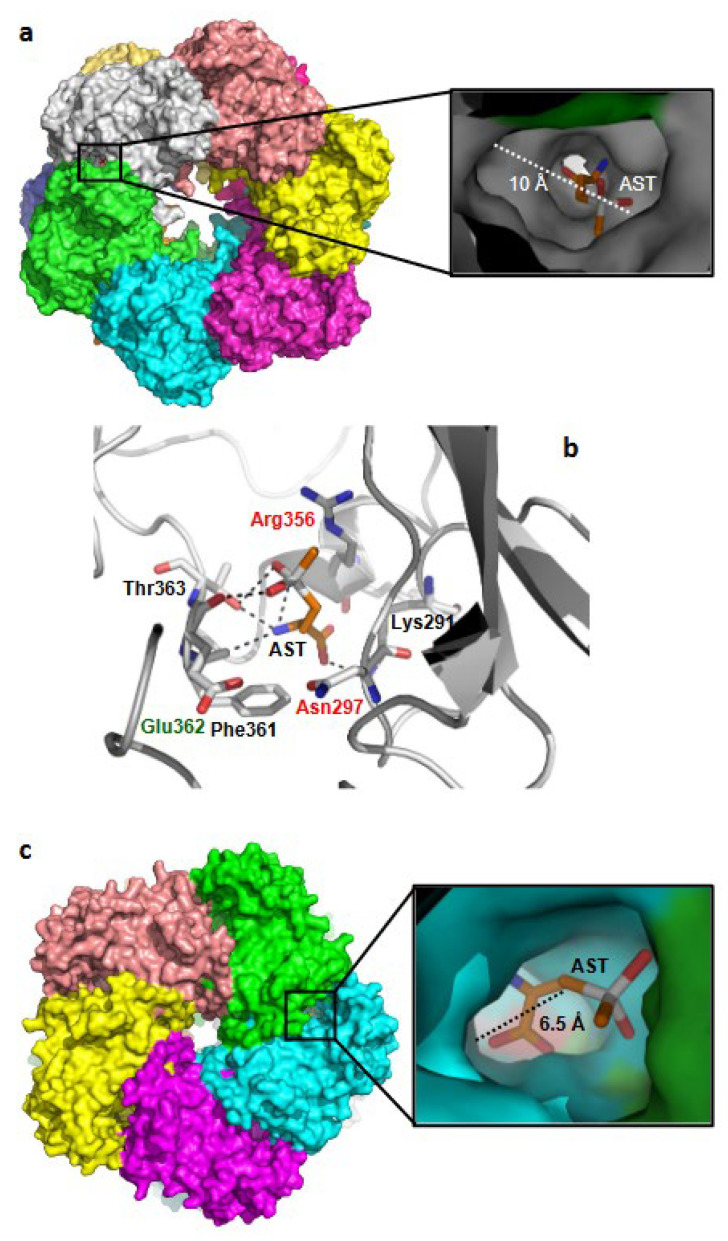
Docking of AST with PfGS-I and hGS-II. (**a**) Surface diagram of PfGS-I docked with AST. PfGS-I is a dodecamer composed of two hexameric rings with 12 active sites, each of which is formed between two monomers. AST molecule is docked with one of the glutamate-binding sites using AutoDock4. The approximate cavity width across the glutamate-binding site is 10 Å. (**b**) Interaction of AST with PfGS-I. Residues predicted to be involved in AST-binding are shown in stick, where residues involved in binding of glutamate and ammonia are highlighted in red and green letters, respectively. The dotted lines represent hydrogen bonds. (**c**) Surface diagram of hGS-II docked with AST. hGS-II is a decamer composed of two pentameric rings with 10 active sites, each of which is formed between two monomers. The AST molecule is docked with one of the glutamate-binding sites using AutoDock4. The approximate width of the cavity of the glutamate binding site is 6.5 Å.

## Data Availability

The data presented in this study are contained within the article and available on request from the corresponding authors.

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
