# Peer review of "Arsinothricin Inhibits Plasmodium falciparum Proliferation in Blood and Blocks Parasite Transmission to Mosquitoes"

_microorganisms, 2023, doi:10.3390/microorganisms11051195_

Round 1

Reviewer 1 Report

Comments on ”Arsinothricin inhibits Plasmodium falciparum proliferation in blood and blocks parasite transmission to mosquitoes”

The manuscript from Yoshinaga and colleagues describes experiments with Arsinothricin, an arsenic compound apparently active against the type Glutamine synthetase. The data are well presented and so far support the major findings. Before publication, I would strongly suggest addressing the following issues.

Please run an English language check, especially in the abstract. While long stretches of text are error-free, others have weird expressions or missing articles.

In the introduction section, why do you emphasize natural compounds (line 51) and not compounds as a whole? In this sense, you may want to see (and cite) the publication by Flaminia Cateruccia’s group regarding the effective killing of Plasmodium inside the mosquito by Atovaquone-sprayed bednets (Nature. 2019 Mar;567(7747):239-243).

The experimental section is accurately written. In line 228, and for the reader non-familiar with the rearing of mosquitoes, please inform which sugar in what liquid was offered to the mosquitoes after the blood meal. In line 234, you inform that you used 300,000 cells per well in 96-well plates. Please check this value, this cell number is very high for such a small well.

In line 345, you state that AST has a moderate anti-asexual activity which sounds awkward, please change to “AST has moderate activity against asexual blood stage forms”.

In line 349, the legend of Figure 3, change “parasitemia rates” for parasitemia. By the way, how did you measure parasitemias. It is stated in line 215/216 that parasitemias were recorded, but how was that done? Via microscopy, cytometry or SYBR based methods?

In Figure 4, the effect of AST on oocyst formation is shown. I can see that the highest concentration of AST is effective against oocyst formation, but does that concentration also kills mosquitoes, given that only 18 mosquitoes were analyzed here? Can you comment on the survival of mosquitoes when being fed with AST?

In line 372 title of this paragraph: please correct. Probably you may want to write:  AST shows low toxicity in human cell lines. Please also comment on using iAs(III) in this context – the figure contains data but you do not mention them in the main text nor in the figure legend.

In line 410 you argue that AST is active against merozoites since it does not seem to go inside erythrocytes regardless of being infected or not. This is an interesting point and would indicate that merozoite PfGS-1 is easily accessed by AST which then would somehow block the invasion process at some point. This is worth checking by an additional assay where E64-blocked purified schizonts have E64 washed out. After that, egressing merozoites would be left to reinvade in the presence of fresh blood. After 12 h incubation, parasitemia would be tested. Although this does not solve the question at which point of the reinvasion process AST may work it would at least underscore the effect against merozoite forms.

The discussion part shows the results of docking studies. The authors should state whether attempts were done to co-crystallize AST with purified recombinant PfGS-I to support the docking studies.

Reviewer 2 Report

The manuscript is appropriate for acceptance with edits. These results will greatly contribute to the field of anti-malarial drug development, 1) by providing a novel molecule that has an effect on Plasmodium falciparum infectivity and transmission, 2) by providing novel drug targets in the urea cycle and 3) by providing the described pipeline and approach which can be used to identify novel drug targets and validating efficacious and feasible molecules.

Major points/questions:

1.     In this manuscript through multiple results we evidently see the inhibition of parasitemia and transmission by AST, but we see evidence of cytotoxicity and arsenic concentrations in human cell lines as well. It is important to discuss the possible harmful effects and provide some comparisons and thresholds of these effects (Based on literature only).

Minor points/suggestions:

1.     General,

a.     In Figures, it will be better to stick to one notation of p-values, either the values itself or *, to keep it consistent throughout the manuscript.

2.     In Introduction section,

a.     In paragraph 1,

                                               i.     Malaria is represented by both P. falciparum and P. vivax worldwide, it will be appropriate to mention that out of the 5 Plasmodium species infecting humans, P. falciparum is responsible for majority of the morbidity and mortality worldwide.

b.     In paragraph 4,

                                               i.     It is not clear how AST inhibits sexual-stage P. falciparum transmission to mosquitoes. Is it because there are eventually less Ring stage parasites (because of reduced Merozoites) available for gametocytogenesis?

                                             ii.     The harmful effects to human host should be discussed.

3.     In Materials and Methods section,

a.     In Section 2.4,

                                               i.     It is not clear from where the genetic sequences were obtained.

                                             ii.     It is important to mention the strain of each species used (Ex. P. vivax Sal-1 or P01?)

                                            iii.     Provide a working link for BoxShade tool.

4.     In Results section,

a.     In Fig. 2 legend,

                                               i.     In last line, Fig. 3b and c are mentioned. It should be 2b and c.

b.     In Section 3.3,

                                               i.     There is some inhibition (~20%) hGS-II observed in Fig. 2b.

c.     In Section 3.4,

                                               i.     Mention the references for transcriptomics and proteomics data. From which study these values are made available on PlasmoDB?

d.     In Section 3.5,

                                               i.     In Fig. 4, where it is shown that AST completely inhibits Malaria transmission, is this because AST acts on Merozoites and reduces the concentration of Merozoites completely that there are no downstream life-cycle stages available leading to gametocytogenesis? Is there an account of different P. falciparum life-cycle stages observed? 

e.     In Fig. 6,

                                               i.     The arsenic levels in all the cell lines are high. Can this have any effect?

5.     In Discussion section,

a.     In paragraph 1, it is important to briefly describe the harmful effects of AST on human cell lines.

b.     In paragraph 3, include the references for studies from which the transcriptomic and proteomic data is recovered (within ToxoDB, CryptoDB and PiroplasmaDB).

c.     In paragraph 5, It is not clear how is AST affecting after the parasite egress in the sexual stage (mosquito midgut). Explain briefly.
